# Fast Focus Ultrasound Liver Technique for the Assessment of Cystic Echinococcosis in Sheep

**DOI:** 10.3390/ani11020452

**Published:** 2021-02-09

**Authors:** Giuliano Borriello, Jacopo Guccione, Antonio Di Loria, Antonio Bosco, Paola Pepe, Francesco Prisco, Giuseppe Cringoli, Orlando Paciello, Laura Rinaldi, Paolo Ciaramella

**Affiliations:** Department of Veterinary Medicine and Animal Productions, University of Napoli “Federico II”, via Delpino, 1, 80137 Napoli, Italy; giuliano.borriello@unina.it (G.B.); jacopo.guccione@unina.it (J.G.); antonio.bosco@unina.it (A.B.); paola.pepe@unina.it (P.P.); francesco.prisco@unina.it (F.P.); giuseppe.cringoli@unina.it (G.C.); orlando.paciello@unina.it (O.P.); laura.rinaldi@unina.it (L.R.); paolo.ciaramella@unina.it (P.C.)

**Keywords:** cyst, *Echinococcus granulosus*, hydatidosis, one health, ultrasonography, zoonosis

## Abstract

**Simple Summary:**

Cystic echinococcosis (CE) is a parasitic zoonosis caused by the larval stage of the tapeworm *Echinococcus granulosus.* The liver ultrasonography can be considered as the gold standard for the in vivo diagnosis detection of CE. Nevertheless, control programs against *E. granulosus* are considered long-term actions that require an integrated approach and the high expenditure of time and financial resources. This parasitosis still causes severe economic losses both in human healthcare and in the entire livestock breeding system, with loss in milk, meat, and wool production. Considering the impact of this zoonosis in the modern sheep’s breeding system, a reliable hepatic ultrasound could represent an innovative strategy to control and even eradicate the disease in sheep farms. Based on these considerations, the present study aimed to evaluate a fast-focused technique for hepatic CE detection in different sheep breeds and compare the performance of the latter with another fast-scan (previously evaluated in Sarda sheep breed), the complete ultrasound liver examination, and the anatomopathological examination.

**Abstract:**

A complete ultrasound examination (cUS) of the liver was performed on 172 female sheep and compared to the performance of a fast-focused ultrasound technique to diagnose echinococcal cysts. The scanned area was divided in: HYP (right hypocondrium), zone (Z)1 from HYP to the 11th intercostal space (IS), Z2 (10th–8th IS) and Z3 (7th–5th IS). Contiguous scans were also examined (HYP + Z1, Z1 + Z2, Z2 + Z3). Furthermore, during the procedures, the sheep were divided into three groups according to the body weight: Group (G) 1 (lighter), G2 (medium), and G3 (heavier). Finally, diagnostic outcomes were compared with necropsy findings. cUS obtained the highest values of sensitivity (Se) (91%), Specificity (Sp) (80%), and positive-zones (124/138, 90%), as compared to the other scans. cUS was also characterized by high values of Se and Sp and was able to identify a great number of positive-zones, when sheep were divided by body-weight groups. Similar performances were obtained in G1 by HYP (Se: 91%–Sp: 82%; 18/20, 90% of positive-zones) and HYP + Z1 scans (Se: 91%–Sp: 82; 90% of positive-zones, 18/20). Thus, in lighter breeds, the examination of HYP and HYP + Z1 scan windows could be considered reliable techniques for identifying the infected animals, while in heavier sheep the cUS still represents the best option.

## 1. Introduction

Cystic echinococcosis (CE), also known as hydatid disease or hydatidosis, is a parasitic zoonosis caused by the larval stage (metacestode) of the tapeworm *Echinococcus granulosus*. The life cycle of the parasite includes dogs and other canids as the definitive hosts, while livestock, as well as humans, are intermediate hosts, harboring the hydatid cysts [1]. CE has a worldwide distribution and exhibits the highest prevalence in communities where pastoral activities predominate such as the Mediterranean areas, eastern Europe, the southern tip of South America, several areas of Central Asia, Siberia, and eastern China [2]. This parasite is responsible for severe economic losses both in human healthcare ($763,980,979/year) and in the entire livestock breeding system ($2,51,409,989/year) [3,4]. Indeed, it has a strong impact on sheep farms’ earnings with loss in milk (7–20%), meat (5–20%), and wool (10–40%) production [5]. In this regard, the control and prevention of this parasitosis represent a crucial step to reduce its negative economic impact, but also to prevent the spread of the disease in rural areas [6]. To date, the diagnosis of CE in small ruminants is still mainly based on post-mortem exam. Indeed, several serological tests have been developed using different substrata (i.e., antigens, antibodies, recombinant protein) without acceptable results [7,8,9,10]. The first diagnostic imaging technique for CE detection used in sheep was based on the radiological examination of lungs [11], subsequently, ultrasonography (US) was employed to assess the prevalence of hydatid cysts in the liver of sheep and goats in Turkana, Kenya [12]. During last years, transcutaneous liver ultrasonography can be considered a reliable, simple, and non-invasive tool for the in vivo diagnosis of cystic echinococcosis in sheep [13,14]. Despite of this, there is a lack of guidelines that clearly indicate the ideal type of ultrasound probe and its proper setting to correctly detect the hydatid cysts in the liver of the sheep.

However, this technique examines the internal structure of the cysts and, therefore, it can effectively differentiate the dead cysts from the viable ones as well as determine their stadium according to several classification primary developed for human medicine and then adapted for the ovine species [9,15,16]. Nevertheless, the time necessary to perform the examination often represents a limit of this diagnostic technique, especially when it is employed as a mass screening in large sheep flocks. Dore et al. [13] validated a liver’s ultrasonographic scanning protocol on the Italian Sarda sheep based on a single acoustic window, placed in the right hypochondrial space, without shaving the animals, reporting satisfactory values of sensitivity (Se) and specificity (Sp) (88.7, 75.9%, respectively). In our preliminary study, the same method, when applied to different sheep breeds did not show similar results especially in not shaved animals [17]. Considering the impact of CE in the modern sheep’s breeding system, a reliable fast-focused hepatic US could represent an important part of an integrated control program for the disease in sheep farms, in order to reduce reluctance of the breeders regarding the use of US in the flock for the fear of economic and time losses [18].

The detection of hydatid cysts by means of US could reduce the spread of the disease through the slaughtering or the treatment of the infected animals. To date, the treatment in sheep relies mainly on anthelminthic drugs, such as albendazole and praziquantel [19] whereas the surgical treatment such has the Puncture-Aspiration-Injection-Reaspiration (PAIR) technique, commonly used in human medicine [20] and, experimentally, performed also in this species [21].

Based on the previous considerations, the present study aimed to (i) evaluate a fast-focused technique for hepatic CE detection in different sheep breeds and (ii) compare the performance of the latter with another fast-scan, the complete US liver examination, and the anatomopathological examination.

## 2. Materials and Methods

### 2.1. Animal Selection

One-hundred-seventy-two female sheep of different breeds, scheduled for slaughter and originating from 21 farms, were enrolled between May 2017 and September 2019. The farms were located in southern Italy within an area showing a CE prevalence up to 75% [2,22]. Eligible criteria for farms were a history indicating the presence of positive animals to CE, the availability of pastures shared with wild animals, and the presence of shepherd dogs within the flock. The sheep enrolled were submitted to a complete clinical examination before the US, recording body condition score (BCS) [23] and body weight (kg) through a weighing scale, at the same time. All the procedures were performed by the investigators abided by the common good clinical practices [24] and received an institutional approval by the Ethical Animal Care and Use Committee of University of Naples Federico II (Protocol No. 67990/2015). Furthermore, the farmers and the owner of the slaughterhouses were previously informed about methods used and in agreement with the purposes of the study.

### 2.2. Ultrasound Examination and Fast Scanning Technique

In each animal enrolled, an area between the right hypochondrium and the caudal margin of the homolateral scapula was shaved (Econom 2, Aesculap Suhl GmbH, DE).

The US was performed with a Mylab^®^ Alpha device (Esaote SPA, IT) using a microconvex multifrequency transducer (6–10 MHz) after the removal of the grease and dirt with a solution of denatured alcool, and the application of the ultrasound gel. All animals were in standing position, not sedated, and handled by a single operator. The time needed to perform each US scan was also recorded from the beginning of clipping to end of the exam (minutes). As described by Braun and Hausammann [25], a complete liver ultrasound examination (cUS) was performed starting from the right hypochondrium, proceeding from 12th to 5th intercostal spaces (IS), each one examined from dorsal to ventral using longitudinal and transverse complete scans.

The scan area was divided into three different scan windows: Zone 1 (Z1), from the right hypochondrium to the 11th IS; Zone 2 (Z2) from the 10th IS to the 8th IS; Zone 3 (Z3) from the 7th IS to 5th IS (Figure 1). Moreover, the liver was also examined placing the probe on the most cranial part of the right hypochondrium (HYP) as described by Dore and collaborators [13]. All the hydatid lesions detected were always localized and characterized according to WHO guidelines [9]. Briefly, the WHO guidelines describe the hydatid cyst as:CL: unilocular anechoic cystic lesion with uniform anechoic content. The cyst wall is not visible.CE1: uniformly anechoic cyst with uniform anechoic content or fine internal echoes.CE2: cyst with internal septation, sometimes with a honeycomb appearanceCE3: unilocular cyst that can show the presence of a daughter cyst.CE4: hypoechoic and hyperechoic matrix, the appearance resembles a ball of wool.CE5: cyst with partially or completely calcified wall.

Finally, recorded data were also considered for contiguous scans (HYP + Z1; Z1 + Z2, Z2 + Z3).

### 2.3. Post-Mortem Examination

Within 24 h from US exam, all sheep were slaughtered, and their liver and lungs were collected and transported to the necropsy room of the Department of Veterinary Medicine and Animal Production, of the University of Naples Federico II. Each organ was carefully valued, and hydatid cysts were recognized and classified according to the WHO [9]. The same investigator macroscopically classified all the lesions; light microscopy was also employed in case of diagnostic doubts.

### 2.4. Statistical Analysis

All the data were analyzed by standard descriptive statistics, normality was assessed using Shapiro–Wilk test. Data were expressed as absolute numbers, percentage, ranges, and mean ± standard deviation (SD). Values of Se, Sp, negative predictive values (NPV), positive predictive values (PPV), number and percentage of hydatid cysts detected were calculated for each scan, using the anatomopathological results as gold standard. The observations of each zone (ZONAL scan) were individually recorded and analysed such as those of the contiguous ones (HYP + Z1; Z1 + Z2; Z2 + Z3). A comparison between each scan window was also performed (i.e., cUS vs. HYP; Z1 vs. Z2 + Z3; Z1 + Z2 vs. cUS, etc). To assess the differences in number and percentage of positive zones found by each scan, a Fisher exact test was used. The statistical significance was determined using a Bonferroni correction (*p* = 0.05/7 = 0.007 for the comparison in the entire population; *p* = 0.05/8 = 0.006 for the comparison between the groups). All statistical analyses were performed using a dedicated software (SPSS, Version 17.0, Chicago, IL, USA). The Se, Sp, NPV and PPV were calculated in order to identify the most effective scan window, as follows:Se = [true-positives/(true-positives + false-negatives)];
Sp = [true-negatives/(true-negatives + false-positives)];
NPV = [true-negatives/(true-negatives + false-negatives)];
PPV = [true-positives/(true-positives + false-positives)]

When the post-mortem examination confirmed the presence of at least one hepatic hydatid cyst detected by US, the zone was classified as true-positive, whereas if there were negative findings in both US and post-mortem examination, the zone was classified as true-negative. Instead, a false-positive was defined as a zone resulted positive for CE at US and negative during the post-mortem exam; a false-negative.

## 3. Results

The data obtained were normally distributed. All animals showed a BCS between 2.5 and 3.5 point and the mean age was 7.3 ± 5.1 years. Because of the non-homogenous weight distribution due to different breeds examined, the sheep were divided into three groups according to Bigi and Zanon [26]: G1 [weight ≤ 45 kg (mean: 43.4 ± 3.2 kg): 22/172, 13%], G2 [46 kg ≤ weight ≤ 63 kg (mean: 61.9 ± 1.6 kg): 69/172, 40%)], G3 [weight ≥ 64 (mean: 76.3 ± 7.4 kg): 81/172, 47%)] (Table 1).

Of the 172 animals examined, 80 (46.5%) were positive for hepatic CE at post-mortem examination. The cUS was able to detect the higher number of true-positive animals (73/172, 42%) when compared to the other scans (Table 2).

In all the animals enrolled, the highest values of Se (91%) were observed when the complete US was employed (Table 3 and Table 4), whereas PPV and NPV values resulted in 80% and 91%, respectively. The cUS confirmed high values of Se and Sp also when the different groups were considered (G1: 91%, 82%; G2: 91%, 72%; G3: 92%, 87%). High values of Se (91%) and Sp (82%) were also observed with HYP scan in G1. When paired, the contiguous zones showed an improved value in Se and Sp, especially in G1 for all the scans (HYP + Z1: 91%, 82%; Z1 + Z2: 91%, 82%; Z2 + Z3: 91%, 82%) and in G3 for Z2 + Z3 (81%, 91%) (Table 4).

The percentages of the true-positive-zones detected with the different US scan windows and groups are reported in Figure 2 and Figure 3. Of 138 positive-zones detected through the liver anatomopathological exam, cUS was able to reveal a significantly higher number of positivity (124/138, 90%) when compared to the other scans (*p* ≤ 0.01) (Figure 2). Considering the positive-zone detected at the liver anatomopathological exam, the G1 showed a significantly higher percentage (18/20, 90%, 95% CI = 72.0–98.0) of positive-zones than the other groups for HYP scan (G2: 28/54, 52%, 95% CI = 41.0–69.0; G3: 24/64, 38%, 95% CI = 28.5–53.0, G1 + G2 + G3: 74/138, 51%, 95% CI = 46.7–62.2) (*p* ≤ 0.01) (Figure 3). Similarly, a significantly higher value (*p* ≤ 0.01) was found in G1 when HYP + Z1 was considered (18/20, 90%, 95% CI = 72.0–98.0) (Figure 3). The Z2 resulted positive more times than the others (Z1: 38/124, 30.7%, 95% CI = 20.6–36.0; Z2: 51/124, 41.1%, 95% CI = 29.3–46.1; Z3: 35/124, 28.2%, 95% CI = 18.6–33.9). This observation was also confirmed by anatomopathological exam (Z1: 41/138, 29.7%, 95% CI = 22.2–38.1; Z2: 53/138, 38.4%, 95% CI = 33.7–50.2; Z3: 44/138, 31.9%, 95% CI = 24.2–40.3).

The mean cUS time needed for the evaluation of a single animal was of 7.1 ± 1.6 min while less time was spent to perform the single scans: HYP: 3.7 ± 0.7, Z1: 1.3 ± 0.6, Z2: 1.3 ± 0.6; Z3, 1.3 ± 0.6 min. Approximately 60% of this time was spent to shave each sheep and prepare the scan area. Data regarding cysts detected by US exam and anatomopathological exam are reported in detail in Table 5. Briefly, a total of 301 cysts (3.8 ± 2.4 cyst each animal) were found in the liver and most of them were classified as inactive (CE5) (Figure 4).

Post-mortem lung hydatid lesions were associated with hydatid liver lesions in 63/80 animals (78.7%); in three animals (3/172, 1.7%) hydatid lesions were present only in the lung parenchyma. Finally, signs of trematodes lesion, as hyperechoic areas (linear or spot) irregularly distributed in the liver parenchyma (ductal phase), were detected both in positive (25/80, 31.2%) and negative (27/92, 29.3%) CE animals. Finally, liver abscesses were detected in 2 sheep (Figure 4).

## 4. Discussion

For the purposes of the analysis outlined in this manuscript, the Se and Sp of the US for hepatic CE detection was performed in different sheep breeds in an endemic area for CE. The estimation of the reliability of this technique against the hydatid cysts represents an interesting challenge in species like sheep, where this parasitosis is common but the in vivo diagnostic tools are quite poor. In our study condition, the complete ultrasound examination (cUS) of the liver was confirmed as a reliable diagnostic method to assess CE infection in alive sheep. Indeed, the investigation revealed how the cUS scan failed in only 7 cases; in all groups considered, Se and Sp values are similar to those reported by Dore et al. [13] (88.7% and 75.9%, respectively), and Hussein et al. [14] (80% and 100%, respectively). The other scan windows employed, even if they failed to detect an acceptable number of positive animals (Se range: 44–78%), could efficiently assess sheep negative for liver CE due to a higher Sp (Sp range: 84–95%) when the entire population was considered (Table 3 and Table 4). Despite the small number of animals involved, only in G1 the HYP and the paired scans reached values of Se and Sp similar to cUS. These results observed only in smaller animals are like those described by Dore et al. [13] who performed the HYP scan, without shaving the scan area in Sarda sheep, an Italian milk breed with a mean weight of 40 kg (female sheep) [26]. However, the significantly higher number of positivity found by HYP (18/20, 90%, 95% CI = 72.0–98.0) and HYP + Z1(18/20, 90%, 95% CI = 72.0–98.0) scans in G1 (*p* ≤ 0.01), compared to the other groups, suggests that these two scans may be efficient in animals weighing less than 45 kg. Indeed, the HYP and the HYP + Z1 scans could effectively scan a large part of the liver parenchyma in these small-sized sheep and, therefore, could better detect the hydatid cyst. When large size sheep were considered, these scans were not able to reach acceptable values of Se, therefore the presence of different breeds could represent a major variable to the correct use of the US in field conditions. This observation does not appear to be influenced by the BCS condition; indeed, the BCS values recorded (2.5–3.5) can be considered in the normal range for small ruminants and related to a little-medium-fat coverage [24]. Moreover, as reported by van Burgel et al. [27], in sheep with BCS values between 2.5 and 3.5 the tissue depth, measured at the 12th intercostal space ranges approximately from 3 mm to 10 mm, and it is not likely to interfere with the ultrasound scanning. The anatomopathological and ultrasonographic analysis of the positivity distribution revealed a predominant concentration in none of the zones considered. In all the examined sheep, lower and not acceptable Se values were obtained either with all the single (Z1: 48%, 95% CI = 36.0–59.0; Z2: 64%, 95% CI = 52.0–74.0; Z3: 44%, 95% CI = 33.0–55.0) or paired scans performed (HYP + Z1: 61%, 95% CI = 50.0–72.0; Z1 + Z2: 78%, 95% CI = 67.0–78.0; Z2 + Z3: 81%, 95% CI = 71.0–89.0) (Table 3 and Table 4); therefore, none of them were shown to be suitable for an on-farm flock screening for CE. Moreover, the non-specific distribution of lesions did not allow the development of a real fast scan procedure, since scanning a partial liver’s areas may easily lead to false-negative results (Table 2). Even in the zonal scan some important differences were recorded when different groups were considered. Indeed, in G3 the large size of the sheep did not allow a reliable scan from the most caudal windows. Based on the previous statement, in this group, Z2 and Z3 showed higher values of Se when compared with the HYP and Z1. Similarly, the low Se value obtained in this heavier group regarding Z1 + Z2 could be the direct consequence of the reasonable difficulties due to the scanning either of the wider thorax or of bowel interposition during the caudal scansion. Although a low number of positive animals for active cysts were found in this study (2/80, 2.5%), in each zone considered and with all the US scan windows used, the Se improved significantly (100%) when active hydatid cysts (CE2) were considered. Therefore, the US may have an important role in the control of *E. granulosus* lifecycle in a flock since this technique reach its maximum efficiency in the diagnosis of active cysts, probably fertile and capable of infection [9]. However, the majority of hepatic cysts detected by US (259/301, 86%), revealed how most of them was inactive (CE4–CE5), suggesting a chronic status of the lesions. Instead, active (CE2) and transitional (CE3) ones were probably a consequence of infection occurred in more recent times. The allocation of *E. granulosus* cysts to their respective ultrasound class (CE1-5) as well as their presence was confirmed by cyst morphology at *post-mortem* examination. As for other studies, focused on the evaluation of the ultrasound liver technique for the diagnosis of CE in sheep [8,13,14], no species-specific PCRs were performed to confirm the presence of *E. granulosus* cysts. However, previous molecular studies on CE in ruminants in the study area confirmed the diagnosis based on cyst morphology and revealed the presence of the *E. granulosus s.s*. genotypes G1 and G3 [22]. Apart from hydatid cysts, US allowed for the detection of further lesions with a different origin; most of them were indeed related to liver flukes both in positive (25/80, 31%) and negative (27/92, 29%) animals. Furthermore, in one sheep a multiple lesion showed a US appearance indicative of both active and inactive cysts although post-mortem exam revealed the presence of multiple abscesses. The presence of these lesions probably originated from complicated lesions due to hydatids, flukes or other pyogenic infection [28] could lead to a misdiagnosis, thus representing a potential limit of US employment for in-vivo diagnosis under field condition. Regarding the simultaneous presence of CE in liver and lungs, it was frequently observed at post-mortem examination (63/80, 79% of cases) similarly to what reported in the literature [13]. The cases in which the lungs were instead the only target organs of CE infection were considered a sporadic event (3/172, 1.7%); this may represent a limit of the exclusive liver US examination. Indeed, if on one side, the only focusing on this organ may sensibly reduce time-consuming for US exam under field condition and may give encouraging results as screening test for positive animals’ detection, on the other side, the lack of information regarding the lung’s status may lead to underestimating the disease’s prevalence within the flock, thus limiting the effectiveness of strategies for prevention and control. Furthermore, under the practical point of view, the preparation (handling and shaving) and the execution of the cUS did not require a large amount of time for a single animal but, when the procedures are applied to an entire flock, it may require a sensible increase of time potentially limiting the use of this diagnostic approach as precision farming technique. However, because most of the time was spent for wool shaving (approximately 60%) the authors suggest the use of US during or immediately after the shearing season.

## 5. Conclusions

The current study represents the first investigation evaluating the use of the US as a potential fast focused technique for CE hepatic lesions detection, in different sheep breeds under field conditions. The investigation confirmed that the ultrasonography can be considered as a reliable intra-vitam technique for assessment of hydatid cysts in the liver. In small breeds (≤45 kg) the use of HYP and HYP + Z1 scan windows can be considered a reliable fast-focused technique for identifying the infected animals, while in heavier sheep (>45 kg) the scan of the entire organ, from the hypochondrium to the 5th IS (Z1→Z2→Z3) was necessary to perform the CE diagnosis. The time needed for the exam execution can represent a limit especially for mass screening in large flocks; further strategies to reduce the time consuming under field condition should be evaluated to improve the widespread of ultrasound use for diagnosis of hepatic CE in sheep. Nevertheless, control programs against *E. granulosus* are considered long-term actions which require an integrated approach and high expenditure of time and financial resources; therefore, the employ of US technique for hepatic CE detection under field conditions can be included in an integrated control strategy in animal and human populations in endemic areas.

## Figures and Tables

**Figure 1 animals-11-00452-f001:**
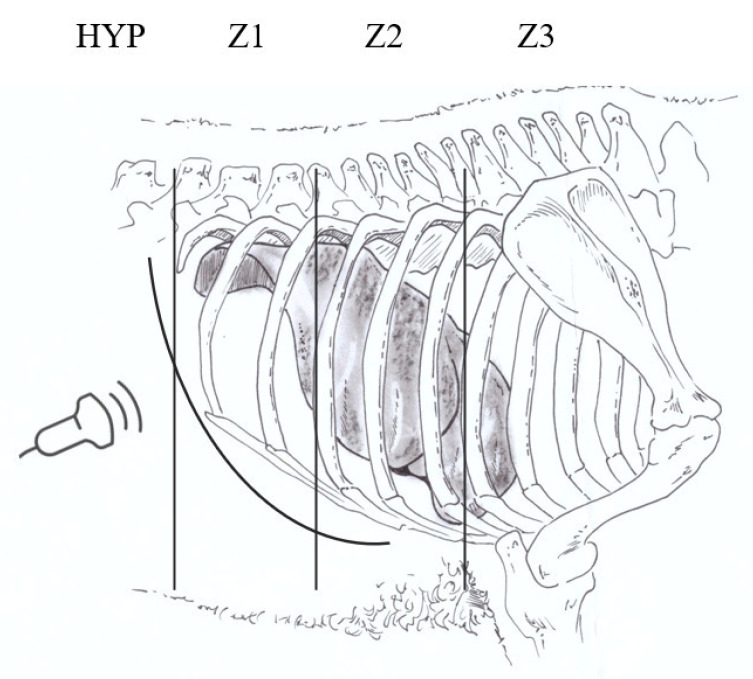
Liver area division for the development of a fast-scanning procedure. HYP: liver scan through hypochondrium; Z1: liver scan from hypochondrium to 11th intercostal space; Z2: liver scan from 10th to 8th intercostal space; Z3: liver scan from 7th to 5th intercostal space.

**Figure 2 animals-11-00452-f002:**
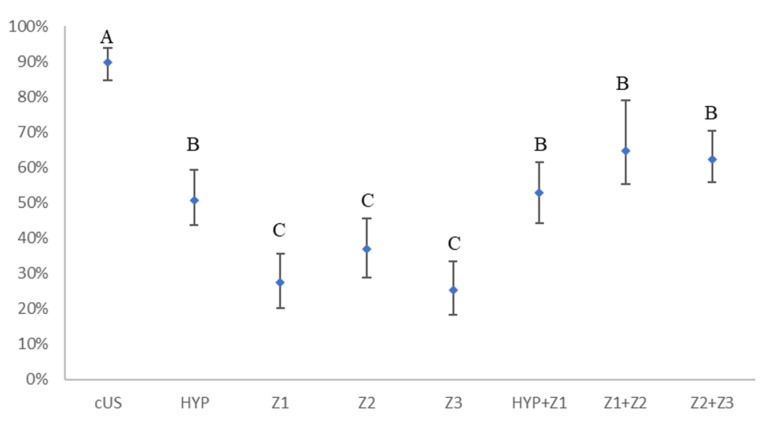
Percentage of true-positive-zones detected with each US scan windows in the entire population. cUS: complete liver scan through intercostal spaces; HYP: liver scan through hypochondrium; Z1: liver scan from hypochondrium to 11th intercostal space; Z2: liver scan from 10th to 8th intercostal space; Z3: liver scan from 7th to 5th intercostal space, HYP + Z1: sum of scan in HYP and Z1; Z1 + Z2: sum of scan in Z1 and Z2; Z2 + Z3: sum of scan in Z2 and Z3; A, B, C: percentage with the same letter are not significantly different (*p* ≤ 0.01).

**Figure 3 animals-11-00452-f003:**
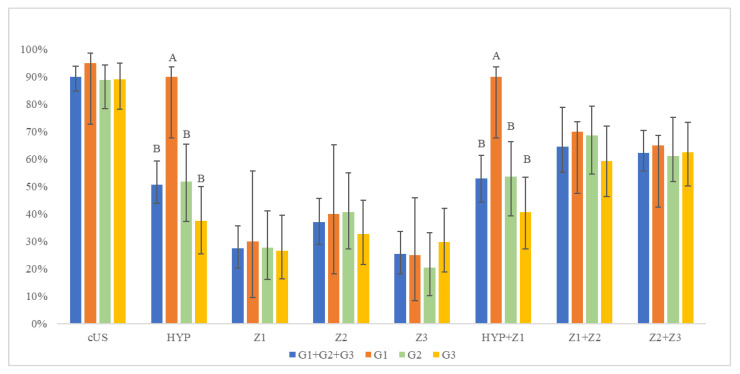
Percentage of true-positive-zones detected with each scan windows in the entire population and groups. cUS: complete liver scan through intercostal spaces; HYP: liver scan through hypochondrium; Z1: liver scan from hypochondrium to 11th intercostal space; Z2: liver scan from 10th to 8th intercostal space; Z3: liver scan from 7th to 5th intercostal space; HYP + Z1: sum of scan in HYP and Z1; Z1 + Z2: sum of scan in Z1 and Z2; Z2 + Z3: sum of scan in Z2 and Z3; G1: weight ≤ 45 kg; G2: 46 kg ≤ weight ≤ 63 kg; G3: weight ≥ 64 kg; G1 + G2 + G3: total population; A, B: percentage with the same letter are not significantly different (*p* ≤ 0.01).

**Figure 4 animals-11-00452-f004:**
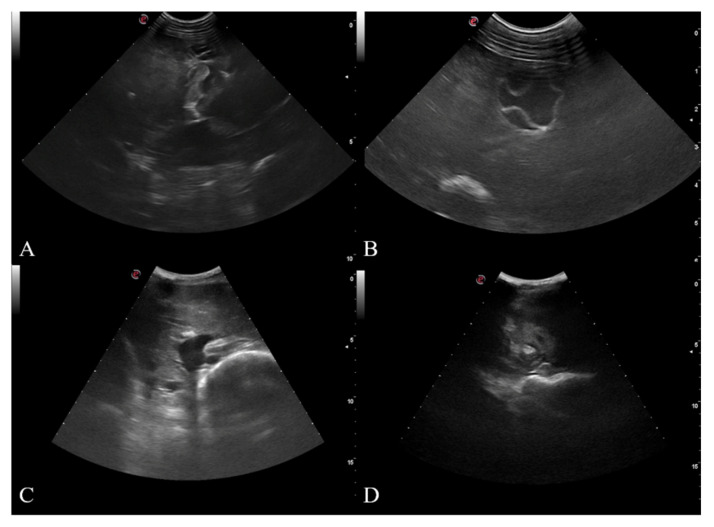
Liver lesion detected through US: (**A**) CE2, multivesicular, multiseptated cysts, active; (**B**) CE3, unilocular cyst, with detachment of laminated membrane from cyst wall, could contain daughter cysts, transitional; (**C**) CE5, thick calcified wall that is arch shaped, producing a cone shaped shadow, inactive; (**D**) liver abscess.

**Table 1 animals-11-00452-t001:** Breeds, number of sheep for each breed (%), and the mean weight (±SD) of examined sheep.

Breed	n° Sheep (%)	Mean Weight (±SD)	G1	G2	G3
Mixed	105/172 (61%)	68.4 ± 9.9	5/22 (23%)	45/69 (65%)	55/81 (68%)
Merinizzata	24/172 (14%)	79.2 ± 2.8	0/22 (0%)	0 (0%)	24/81 (30%)
Bagnolese	21/172 (12%)	62.0 ± 2.5	0/22 (0%)	19/69 (28%)	2/81 (2%)
Comisana	10/172(6%)	51.5 ± 7.6	5/22 (23%)	5/69 (7%)	0 (0%)
Sarda	12/172 (7%)	42.1 ± 4.0	12/22 (55%)	0 (0%)	0 (0%)

n°: number of sheep; SD: standard deviation; G1: weight ≤ 45 kg; G2: 46 kg ≤ weight ≤ 63 kg; G3: weight ≥ 64 kg; G1 + G2 + G3: total population.

**Table 2 animals-11-00452-t002:** Ultrasonography findings with each scan windows for each group.

		cUS	HYP	Z1	Z2	Z3	HYP + Z1	Z1 + Z2	Z2 + Z3
G1 + G2 + G3	TP	73	48	38	51	35	49	62	65
		(42%)	(28%)	(22%)	(30%)	(20%)	(28%)	(36%)	(37%)
	FP	18	8	5	11	5	8	15	15
		(10%)	(5%)	(3%)	(6%)	(3%)	(5%)	(9%)	(9%)
	TN	74	84	87	81	87	84	77	77
		(43%)	(49%)	(51%)	(47%)	(51%)	(49%)	(45%)	(45%)
	FN	7	32	42	29	45	31	18	15
		(4%)	(18%)	(24%)	(17%)	(26%)	(18%)	(10%)	(9%)
	TOTAL	172	172	172	172	172	172	172	172
		(100%)	(100%)	(100%)	(100%)	(100%)	(100%)	(100%)	(100%)
G1	TP	10	10	6	8	5	10	10	10
		(45%)	(45%)	(27%)	(36%)	(23%)	(45%)	(45%)	(45%)
	FP	2	2	1	2	1	2	2	2
		(9%)	(9%)	(5%)	(9%)	(5%)	(9%)	(9%)	(9%)
	TN	9	9	10	9	10	9	9	9
		(41%)	(41%)	(45%)	(41%)	(45%)	(41%)	(41%)	(41%)
	FN	1	1	5	3	6	1	1	1
		(5%)	(5%)	(23%)	(14%)	(27%)	(5%)	(5%)	(5%)
	TOTAL	22	22	22	22	22	22	22	22
		(100%)	(100%)	(100%)	(100%)	(100%)	(100%)	(100%)	(100%)
G2	TP	30	20	15	22	11	21	27	26
		(43%)	(29%)	(22%)	(32%)	(16%)	(31%)	(39%)	(38%)
	FP	10	5	3	6	2	5	9	8
		(15%)	(7%)	(4%)	(9%)	(3%)	(7%)	(13%)	(11%)
	TN	26	31	33	30	34	31	27	28
		(38%)	(45%)	(48%)	(43%)	(49%)	(45%)	(39%)	(41%)
	FN	3	13	18	11	22	12	6	7
		(4%)	(19%)	(26%)	(16%)	(32%)	(17%)	(9%)	(10%)
	TOTAL	69	69	69	69	69	69	69	69
		(100%)	(100%)	(100%)	(100%)	(100%)	(100%)	(100%)	(100%)
G3	TP	33	18	17	21	19	18	25	29
		(41%)	(22%)	(21%)	(26%)	(24%)	(22%)	(31%)	(36%)
	FP	6	1	1	3	2	1	4	5
		(7%)	(1%)	(1%)	(4%)	(2%)	(1%)	(5%)	(6%)
	TN	39	44	44	42	43	44	41	40
		(48%)	(55%)	(54%)	(52%)	(53%)	(55%)	(51%)	(50%)
	FN	3	18	19	15	17	18	11	7
		(4%)	(22%)	(24%)	(18%)	(21%)	(22%)	(13%)	(8%)
	TOTAL	81	81	81	81	81	81	81	81
		(100%)	(100%)	(100%)	(100%)	(100%)	(100%)	(100%)	(100%)

cUS: complete liver scan through intercostal spaces; HYP: liver scan through hypochondrium; Z1: liver scan from hypochondrium to 11th intercostal space; Z2: liver scan from 10th to 8th intercostal space; Z3: liver scan from 7th to 5th intercostal space; HYP + Z1: sum of scan in HYP and Z1; Z1 + Z2: sum of scan in Z1 and Z2; Z2 + Z3: sum of scan in Z2 and Z3; TP: true positive; FP: false positive; TN: true negative; FN: false negative; G1: weight ≤ 45 kg; G2: 46 kg ≤ weight ≤ 63 kg; G3: weight ≥ 64 kg; G1 + G2 + G3: total population.

**Table 3 animals-11-00452-t003:** Values of sensitivity and specificity of ultrasonography (US) in the different groups.

	cUS		HYP		Z1		Z2		Z3	
	Se(%)	Sp(%)	Se(%)	Sp(%)	Se(%)	Sp(%)	Se(%)	Sp(%)	Se(%)	Sp(%)
	(95% CI)	(95% CI)	(95% CI)	(95% CI)	(95% CI)	(95% CI)	(95% CI)	(95% CI)	(95% CI)	(95% CI)
G1	91 (59–100)	82 (48–98)	91 (59–100)	82 (48–98)	55 (23–83)	91 (59–100)	73 (39–94)	82 (48–98)	45 (17–77)	91 (59–100)
G2	91 (76–98)	72 (55–86)	61 (42–77)	86 (71–95)	47 (28–64)	92 (78–98)	67 (48–82)	83 (67–94)	33 (18–52)	94 (81–98)
G3	92 (78–98)	87 (73–95)	50 (33–67)	98 (88–100)	47 (30–65)	98 (88–100)	58 (41–74)	93 (82–99)	53 (35–70)	96 (85–98)
G1 + G2 + G3	91 (81–96)	80 (71–88)	60 (48–71)	91 (84–96)	48 (36–59)	95 (88–98)	64 (52–74)	88 (80–84)	44 (33–55)	95 (88–98)

cUS: complete liver scan through intercostal spaces; HYP: liver scan through hypochondrium; Z1: liver scan from hypochondrium to 11th intercostal space; Z2: liver scan from 10th to 8th intercostal space; Z3: liver scan from 7th to 5th intercostal space; Se: sensitivity; Sp: specificity; G1: weight ≤ 45 kg; G2: 46 kg ≤ weight ≤ 63 kg; G3: weight ≥ 64 kg; G1 + G2 + G3: total population.

**Table 4 animals-11-00452-t004:** Values of sensitivity and specificity of scan of paired zones for each group.

	HYP + Z1		Z1 + Z2		Z2 + Z3	
	Se (%)	Sp (%)	Se (%)	Sp (%)	Se (%)	Sp (%)
	(95% CI)	(95% CI)	(95% CI)	(95% CI)	(95% CI)	(95% CI)
G1	91 (59–100)	82 (48–98)	91 (59–100)	82 (48–98)	91 (59–100)	82 (48–98)
G2	64 (45–80)	86 (71–95)	82 (65–93)	75 (58–88)	79 (61–91)	81 (61–80)
G3	50 (33–67)	98 (88–100)	69 (52–84)	91 (79–98)	81 (64–92)	89 (76–96)
G1 + G2 + G3	61 (50–72)	91 (84–96)	78 (67–86)	84 (75–91)	79 (71–89)	84 (75–91)

HYP + Z1: sum of scan in HYP and Z1; Z1 + Z2: sum of scan in Z1 and Z2; Z2 + Z3: sum of scan in Z2 and Z3; Se: sensitivity; Sp: specificity; G1: weight ≤ 45 kg; G2: 46 kg ≤ weight ≤ 63 kg; G3: weight ≥ 64 kg; G1 + G2 + G3: total population.

**Table 5 animals-11-00452-t005:** Hydatid cysts detected through ultrasonographic and anatomopathological examination of the liver.

	cUS	AE
CL	0% (0/259)	0% (0/301)
CE1	0% (0/259)	0% (0/301)
CE2	1.2% (3/259)	1% (3/301)
CE3	5% (13/259)	4.6% (14/301)
CE4	1.5% (4/259)	3% (9/301)
CE5	92.3% (239/259)	91.4% (275/301)

cUS: complete liver scan through intercostal spaces; AE: anatomopathological examination; CL: unilocular cystic lesion; CE1: unilocular, simple cyst with uniform anechoic content (Active); CE2: multivesicular, multiseptated cysts (Active); CE3: unilocular cyst which may contain daughter cysts (Transitional); CE4: heterogenous, hypoechoic or hyperechoic degenerative cyst, without daughter cysts (Inactive); CE5: cyst characterized by thick calcified wall, producing a cone shaped shadow (Inactive).

## Data Availability

All data used in the current study are available from the corresponding author on reasonable request.

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
