# Peer review of "Fast Focus Ultrasound Liver Technique for the Assessment of Cystic Echinococcosis in Sheep"

_animals, 2021, doi:10.3390/ani11020452_

Round 1

Reviewer 1 Report

This is a very interesting and clinical useful manuscript about the use of fast focused ultrasound liver technique for the assessment of CE in sheep. As the author stated, a reliable fast-focused hepatic US could represent an important step to control and even eradicate the disease in sheep farms. Moreover, it is a well-written and designed paper with a significant potential use in clinical practice

I only found the following typos:

Lines 295-299: the sentence “However, because most of the time was spent for wool shaving (approximately 60%) the authors suggest the use of this diagnostic tool during or immediately after the shearing season” is repeated.

Line 117: “Shapiro Wilk tests” should be replaced by “Shapiro Wilk test”

I suggest authors to replace “cUS: liver scan through intercostal spaces” with “cUS: complete liver scan through intercostal spaces” or with “cUS: complete liver scan" in lines 149, 164, 185, 193 and 207.

Author Response

The authors warmly thank the reviewer for their valuable and interesting comments and suggestions.

Reviewer 2 Report

This study includes very interesting contents, enough for publication in “Animals”. However, this study is written by very incomplete and poor descriptions, especially methodology (although it is study concept).

The authors did not make this report using all of the previous relevant reports.

At least, the following previous ovine reports should be used as references in the revised version.

  • Maxson AD, Wachira TM, Zeyhle EE, Fine A, Mwangi TW, Smith G. The use of ultrasound to study the prevalence of hydatid cysts in the right lung and liver of sheep and goats in the Turkana. International Journal for Parasitology 1996;26:1335–1338.
  • Lahmar S, Ben Chéhida F, Pétavy AF, Hammou A, Lahmar J, Ghannay A, Gharbi HA, Sarciron ME. Ultrasonographicscreening for cystic echinococcosis in sheep in Tunisia. Veterinary Parasitology 2007;143:42–49.
  • Guarnera EA, Zanzottera EM, Pereyra H, Franco AJ. Ultrasonographic diagnosis of ovine cystic echinococcosis. Vet Radiol Ultrasound 2001;42:352-354.
  • Njoroge E Thesis. “Evaluation of ultrasonographic diagnosis, treatment methods and epidemiology of cystic echinococcosis in sheep and goats.”

In Njoroge E Thesis, I recommend the reference use of the following contents:

  • Pages 135-144: CHAPTER 8 Evaluation of ethyl alcohol in treatment of cystic echinococcosis using puncture, aspiration, introduction, reaspiration (PAIR) technique. Based on the description in this paragraph, the ultrasonography-guided liver centesis can be utilized as the diagnostic method?
  • Pages 33-40: Please compare between the classification results of this study and the Gharbi classification, which is classified in five types.

(Gharbi HA, Hassine W, Brauner, MW, Dupuch K. Ultrasound examination of hydatid liver. Radiology 1981;139:459-463)

The descriptions of this report should be corrected or added according to the following advices:

The ultrasonographic classification of the hepatic cysts according to WHO guideline is applicable to sheep? Applicable because livestock as well as humans are intermediate hosts? Is it different in the development time of the formation of the hepatic cysts between human and sheep?

The author should describe the comparison between human and ovine cases in the percentage of each WHO classification (CL, CE1 to CE5). Why >90% of sheep had CE5 lesions? The proportion of CE5 is >90% in human patients?

How much was the average number of the cystic lesions per one animal?

How much was the ages of the examined sheep? The older sheep have more chronic lesions rather than the younger sheep?

Based on the data shown in Table 2, the order of scanning can be recommended “Z3”, ”Z2”, ”Z1” and “HYP” in the larger sheep? It is because the percentage was larger “Z2+Z3” rather than “Z1+Z2” and “Z1 and HYP’”. Otherwise, is it recommended to start scanning at the area of “Z2”, based on the data shown in Table 3? It is interesting for me to show the recommended order of the area scanned.

Please provide the data (such as true positive, false positive, true negative, false negative, NPV, PPV, examination time, etc.) in each five breeds. The largest Merinizzata breed had poorer ultrasonographic performance than the smallest Sarda breed? In addition, the “HYP” or “HYP+Z1” scans are more applicable to the Sarda breed?

Are there the difference among five breeds in the quality (soft, hard and so on) and quantity (few, many, sparse and so on) of the hairs? The sentence in lines 239-240 notes that the scanning was possible without shaving in Sarda sheep, Scanning without shaving is possible in the other breeds examined in this study?

Line 28: “Se” should be replaced by “sensitivity (Se)”, and “Sp” should be replaced by “specificity (Sp)”.

Line 54: “Gold standard” is liver’s ultrasonography in diagnosing cystic echinococcosis?

Line 72: Why 172 sheep were scheduled for slaughtering? These ovine cases had which types of the clinical signs?

Line 75: “[14,2]” should be replaced by “[2,14]”.

Line 87: Why ultrasound gel and alcohol spray have not been used? If these have been used, clipping is not needed?

Line 88: Use of microconvex transducer is recommended rather than linear transducer? The recommended frequency of ultrasound wave is 6-10MHz? The deeper sites of the lives can be visualized using lower frequency of ultrasound wave (<5 MHz)? You should describe the influences of the function of ultrasonography (frequency, probe-type, different machine and so on) for the visibility of hepatic lesions, by compared with those in the following previous reports:

(1) Vet Rec Open 2014;1:e000004

  Animals: Baladi sheep (aged three to six years).

  Ultrasonography: 3.5, 5.0 and 8.0 MHz linear and convex transducers. Clipped and applied a coupling gel before examination.

  Scanning window: the right 12th to 7th intercostal space.

(2) Vet Parasitol 2014;203;59-64

  Animals: 129 Sarda sheep (aged mean 6.5 years, 2.5-13 years).

  Ultrasonography: 4-11 MHz microconvex transducer. No clipped and applied an alcohol and a coupling gel before examination.

  Evaluation: 1) Lesion location in right lobe, left lobe, caudate lobe, quadrate lobe and hilum; 2) WHO classification.

You should describe comparison between Dore’s data and your data: 1) In Dore’s data, the right and left lobes were the more frequent affected region of the livers. How does this Dore’s result influence your data? Using your method, scanning in each area of HYP, Z1, Z2 and Z3 enables to visualize which of the hepatic lobes (right lobe, left lobe, caudate lobe, quadrate lobe and hilum)? 2) In Dore’s study, CE1 and CE2 lesions could be detected. Why there is great difference between the Dore's data and your data?

(3) Intern J Parasitol 1998;28;349-353

  Animals: 16 sheep (aged >1 years).

  Ultrasonography: 3.5 MHz linear transducer.

  Evaluation: Cyst size.

Please provide the result about the cystic sizes in your study.

(4) Vet Parasitol 2007;143:42–49

  Animals: 1039 sheep (aged 1 and 14 years estimated teeth exam)

  Ultrasonography: Portable machine (SCANNER 100 FALCO VET, Type ESAOTE) for 796 animals; A fixed echograph (General Electric RT X 200 equipment) with 3.5 MHz convex and 7.5 MHz linear transducers. 

  Evaluation: Gharbi classification (Gharbi et al., 1981).

  Please compare between the classification results of your study and the Lahmar’s data (using the Gharbi classification).

(5) International Journal for Parasitology 1996;26:1335–1338

  Animals: 260 sheep.

  Ultrasonography: 3.5 MHz linear transducer. Clipped and applied a coupling gel before examination.

  Evaluation: Cyst size.

Lines 100-101: Each characteristic due to the classification (into CL and 5 types) of WHO guideline should be described shortly here.

Table 1: the number of G1, G2 and G3 should be described for each of 5 breeds.

Lines 145-170: The results should be described in all data, even if there is no significant difference.

Line 178: Why this statistical significance is evaluated by P<0.01, although a statistical significant between HYP and Z1+Z2 is P<0.05 in Figure 2.

Figure legend of Figure 2: I cannot understand the explanation of “A,B,C”. You mean “Data with different superscript letters (A-C) are significantly different (P < 0.01”? In addition, I cannot understand explanation of “a,b”. Is there a significant difference between a and b?

Figure legend of Figure 3: I cannot understand the explanation of “A,B”. The terms “A,a” are used in Figure 3.

Why the significant difference is found between G1 and G2/G3/G1+G2+G3 when scanning in the area of HYP?

Why CE 5 lesion is majority of the hepatic lesion (approximately 90%) in this study?

Lines 218-219: I cannot know whether this description is true or not. Hydatid liver lesion can transform the lung hydatid lesion? Hydatid lung lesion cannot transform the liver hydatid lesion?

Lines 229-230: Please correct from “the complete ultrasound examination (cUS) of the liver” to “cUS”.

Lines 246-252: Over-discussed, although this study did not include BCS data.

Lines 268-271: This finding is obtained from the human patients? This finding is applicable to the ovine cases? The active ultrasonographic pattern has been previously observed in application to the ovine cases?

Line 349: Please correct the page number.

Author Response

(The authors gave the same response as above.)

Reviewer 3 Report

The authors used the fast focus ultrasound liver technique for diagnosing cystic echinococcosis in sheep and validated this method by identifying infected animals only based on cyst morphology after the sheep had been slaughtered. I suggest that the diagnosis based on cyst morphology is confirmed by species-specific PCR. If this is not possible, the issue should at least be discussed in the manuscript.

95% confidence intervals are missing for all sensitivity and specificity estimates and need to be added.

In the statistical analysis of the numbers and percentages of positive zones found in each scan, the Fisher exact test should be used. If I understand the analysis correctly, a correction of the p-value for multiple testing needs to be performed (e.g. Bonferroni or Holm correction).

In the conclusions (lines 308 ff.), the time needed for the examinations is mentioned, although this parameter was not measured or assessed in the study. Please revise or delete.

The entire manuscript needs to be revised by a native speaker of English (incorrect us of genitive, of singular instead of plural etc.; please avoid colloquial language).

Detailed comments

Line 16: Since the life cycle of the E. granulosus is complex, the statement that “hepatic ultrasound could represent an innovative strategy to control and even eradicate the disease in sheep farms” seems exaggerated and inappropriate. Please rephrase this sentence.

Lines 28, 31, 141, 142, 174, 177, 219, 222…: Percentages should not separated by “-“ (minus), this may lead to confusion.

Lines 116-117: Why is the Shapiro Wil test mentioned here? I could not find any result, where this test has been applied. Please delete if not needed or explain.

Lines 163-168: Please add 95% confidence intervals for the Se and Sp estimates.

Author Response

(The authors gave the same response as above.)

Round 2

Reviewer 3 Report

I would like to thank the authors for taking my comments and suggestions into account. In my view, the manuscript has been substantially improved.

Minor comment:

Line 28: Please insert spaces after "sensitivity" and "specificity".